# Excess-entropy scaling in supercooled binary mixtures

Ian H. Bell[1], Jeppe C. Dyre [2] & Trond S. Ingebrigtsen [2✉]

Transport coefficients, such as viscosity or diffusion coefficient, show significant dependence on density or temperature near the glass transition. Although several theories have been proposed for explaining this dynamical slowdown, the origin remains to date elusive. We apply here an excess-entropy scaling strategy using molecular dynamics computer simulations and find a quasiuniversal, almost composition-independent, relation for binary mixtures, extending eight orders of magnitude in viscosity or diffusion coefficient. Metallic alloys are also well captured by this relation. The excess-entropy scaling predicts a quasiuniversal breakdown of the Stokes-Einstein relation between viscosity and diffusion coefficient in the supercooled regime. Additionally, we find evidence that quasiuniversality extends beyond binary mixtures, and that the origin is difficult to explain using existing arguments for single-component quasiuniversality.

[1] Applied Chemicals and Materials Division, National Institute of Standards and Technology, Boulder, CO 80305, USA. [2] Glass and Time, IMFUFA, Department of Science and Environment, Roskilde University, Postbox 260, Roskilde DK-4000, Denmark. ✉email: trond@ruc.dk

Supercooled liquids approaching the glass transition show significant non-Arrhenius temperature or density dependence of their transport coefficients, such as viscosity or diffusion coefficient. Several theories have been proposed to explain this phenomenon, for instance: random first-order transition theory, entropy-controlled theories, dynamical facilitation, bond-orientational order, free-volume theories, elastic models, and more[1–15]. Despite these intriguing theories, a broadly-accepted and universal picture of what controls the change in transport coefficients near the glass transition has not yet manifested itself, even for the simplest supercooled liquids.

Rosenfeld discovered[16,17] in 1977 that transport coefficients in the liquid phase are correlated to the excess entropy $S_{ex}$, where $S_{ex}$ is defined by subtracting the ideal gas contribution from the entropy at the same density $\rho$ and temperature $T$, i.e., $S_{ex}(\rho, T) \equiv S(\rho, T) - S_{id}(\rho, T)$. The excess entropy is a negative quantity since the liquid is more ordered than the ideal gas. Rosenfeld found by applying appropriate dimensionless units that the viscosity and diffusion coefficient collapse to a univariate function of the excess entropy for single-component atomic liquids. Since then, excess-entropy scaling has been the focus of simulation and experimental studies in various contexts[18–39], including atomic mixtures, molecular liquids, ionic liquids, network-forming liquids, nanoconfined liquids, nonlinear sheared liquids, and more. For a recent review of excess-entropy scaling, see ref. [40].

Explanations for excess-entropy scaling have been attempted[18,19] from frameworks such as mode-coupling theory[12]. The fact that excess entropy correlates to transport coefficients may also be explained in the context of R-simple liquids and their isomorphs[41–47]. Isomorphs are curves in the thermodynamic phase diagram along which structure and dynamics are invariant in appropriate dimensionless units. Some thermodynamic quantities are also invariant along isomorphs, e.g., the excess entropy[44,46]. Since the excess entropy and the dynamics in dimensionless units, and hence also dimensionless transport coefficients, are invariant along the same curves, one can write $\widetilde{X} = f(S_{ex})$, where $\widetilde{X}$ is a generic dimensionless transport coefficient and $f$ is some a priori system-specific function not predicted by isomorph theory[44,46].

R-simple liquids are defined in computer simulations by reference to the correlation coefficient $R = \langle \Delta W \Delta U \rangle / \sqrt{\langle (\Delta W)^2 \rangle \langle (\Delta U)^2 \rangle}$ calculated from the virial $W$ and potential energy $U$ constant-volume canonical-ensemble fluctuations at a given state point. A pragmatic definition of this class of liquids is $R \geq 0.90$ which depends on the state point[41]. R-simple liquids include most or all van der Waals and metallic liquids, but exclude network-forming, covalent-bonding, and strongly ionic or dipolar liquids. R-simple liquids have been shown to exist both in experiments and simulations, and the concept is also relevant for the crystal region, under nanoconfinement, in nonlinear shear flow, and more[26,48–54]. A review of R-simple liquids and their isomorphs is given in ref. [55].

Rosenfeld reported in his seminal paper a quasiuniversal relation[16,17] for single-component atomic liquids given by the expression

$$\widetilde{X} \cong A \exp[B S_{ex}/k_B N], \tag{1}$$

in which $k_B$ is Boltzmann's constant, $N$ is the number of particles, and $A$ and $B$ are system-independent constants, with, e.g., $A \approx 0.6$ and $B \approx 0.8$ for diffusion[16] and $A \approx 0.2$ and $B \approx -0.8$ for viscosity[17]. Equation (1) enables prediction of unknown transport coefficients for a given system if its excess entropy is known. Later studies revealed that the exponential behavior of the excess entropy does not apply for supercooled liquids whereas excess-entropy scaling in the form $\widetilde{X} = f(S_{ex})$ may still apply[25,31,34,40]. Furthermore, the quasiuniversal relation of single-component atomic liquids was found to break down for, e.g., molecules which in general do not show quasiuniversal behavior[25,56,57]. The quasiuniversal behavior of single-component atomic liquids may be explained by the exponential (EXP) pair potential[58–60], which can be used as a basis for expanding other pair potentials under certain conditions.

Notwithstanding the importance of excess-entropy scaling for single-component atomic liquids, mixtures of atoms are more often used in simulations and experiments to avoid crystallization and to obtain desirable properties in, e.g., metallic alloys[61]. Alas, for atomic mixtures one does not expect quasiuniversality; mixtures may involve atoms of various sizes, different compositions, alongside different interactions amongst the constitutent particles. Krekelberg et al.[22] found poor scaling with excess entropy for binary hard-sphere (HS) mixtures with respect to composition and size and formulated a generalized excess-entropy scaling to remedy this problem. Banerjee et al.[34] studied supercooled binary mixtures and found no universal collapse between a tetrahedral-forming ionic melt and other simple mixtures. This has also been found for other ionic melts in the supercooled region[62]. On the other hand, Lötgering-Lin et al.[36] found collapse with composition of binary Lennard–Jones (LJ) mixtures in the high-temperature regime over a limited range in viscosity (factor of two). A related result has also been found for the computer-simulated metallic alloy AlNi in the high-temperature limit[32]. On account of the high temperatures simulated these mixtures are expected to behave approximately as single-component atomic liquids, and the results are therefore consistent with the previously mentioned studies and results.

The Stokes–Einstein (SE) relation connects the diffusion coefficient $D$ of a large particle immersed in a solvent with viscosity $\eta$, predicting that $D \propto \eta^{-1} T$. The SE relation breaks down in the supercooled regime and explanations have been presented from various theoretical perspectives[33,38,63–70]. Flenner et al.[68] obtained a good collapse of the diffusion coefficient plotted against the structural relaxation time (which may be used as a proxy for the viscosity) for supercooled binary mixtures by scaling the diffusion coefficient and the relaxation time. In other words, the authors showed that the breakdown of SE in the supercooled regime occurs at the same scaled relaxation time and in a quasiuniversal manner for these binary mixtures. Flenner et al.[68] also showed that dynamical heterogeneity exhibits universal features for supercooled liquids. However, it is not clear why a quasiuniversal curve should be observed in the supercooled region as the binary mixtures are very different, and this was also noted by the authors. The focus of the present study is not on the origin behind the SE breakdown, but on the possible quasiuniversality observation of Flenner et al.[68] related to the SE breakdown.

The above observations motivate us to carry out an in-depth study of viscosity and diffusion coefficient going deep into the supercooled regime of a wide range of binary atomic mixtures to investigate whether quasiuniversality applies to mixtures, contrary to the expectation and findings of previous studies. We use molecular dynamics GPU-based computer simulations in the NVT ensemble (the RUMD package[71]) to study six different binary mixtures: The Kob–Andersen binary Lennard–Jones (KA) mixture, the Wahnström (WS) mixture, the generalized LJ (GLJ) mixture, the KA exponential pair potential (KAEXP) mixture, alloys of copper and zirconium (CuZr), and a size asymmetric (AS) mixture. The systems under study include additive and nonadditive mixtures, different steepness of the pair interactions, effective medium interactions, various size asymmetries, and different compositions. Model and simulation details are found in

the Methods section. The Supplementary Tables S1 and S2 include all simulation results in a tabular form. The virial potential-energy correlation coefficient $R$ is >0.90 at all investigated state points, except for the $CuZr_{36:64}$ and AS mixtures where it is somewhat below 0.90 (see Supplementary Tables S1 and S2); some of these systems have previously been investigated in detail for isomorphs see, e.g., refs. [26,41,44,52–54].

The computer models have various degrees of glass-forming ability and thus different ranges of supercooling. Throughout the study, we use two different sets of dimensionless units: one using the microscopic parameters of the potentials based on the length and energy scales of the larger (A) particle, which is standard in computer simulations, and another set of dimensionless units using macroscopic quantities with length given in units of $\rho^{-1/3}$, energy in units of $k_BT$, and time in units of $\rho^{-1/3}\sqrt{m/k_BT}$ ($m$ is the particle mass), as applied in excess-entropy scaling and the isomorph theory[16,44,59]. The macroscopic dimensionless units are termed reduced units and use a tilde above the variable name; microscopic dimensionless units are implicitly assumed when no tilde is given (an exception is the metallic alloys; see "Methods" for their units).

The main findings of the current study are: (1) A nearly composition-independent excess-entropy scaling relation for all studied binary mixtures extending over eight orders of magnitude in viscosity or diffusion coefficient, going three to four orders of magnitude below the mode-coupling temperature $T_{MCT}$ (i.e., where the dynamics starts to become landscape dominated). (2) A quasiuniversal excess-entropy relation amongst binary atomic mixtures with different interactions (e.g., pair interactions and effective medium interactions), mixing rules, and size asymmetry. We find, additionally, that the departure from universality in the supercooled regime can be rationalized using the so-called density-scaling exponent. As a consequence of these findings, we show that the product of viscosity and diffusion coefficient has virtually the same excess-entropy dependence for all mixtures. Our results thus rationalize the observations of Flenner et al.[68] that SE breaks down at the same scaled relaxation time. The presented simulation results are corroborated by experimental data on metallic alloys from the literature which additionally support the validity of the scalings beyond binary mixtures.

## Results

**Excess-entropy scaling**. The study commences by demonstrating deeply supercooled dynamics exemplifed by the self-part of the intermediate scattering function (ISF; see "Methods" for definition) for the KA mixture at 2:1 composition in Fig. 1a. The value of the wave vector $q$ is that of the first peak of the static structure factor. The 2:1 composition is a much better glass former than the standard 4:1 composition[72–74], thus giving access to a wider dynamical range. The supercooled dynamics goes three to four decades below $T_{MCT}$ where the standard 4:1 composition would crystallize. The mode-coupling temperature for the 2:1 KA mixture is $T_{MCT} = 0.55$. We find a plateau in the ISF extending over almost five decades with a stretching exponent $\beta = 0.55$ at the lowest temperature, i.e., the ISF is well fitted by the stretched exponential function $\exp[-(t/\tau_\alpha)^\beta]$, where $\tau_\alpha$ is the $\alpha$-relaxation time. Figure 1b displays the viscosity $\eta$ as a function of $1/T$ for all the studied binary mixtures and shows in all cases strong deviations from a straight line in the supercooled regime, i.e., a significant non-Arrhenius behavior.

Figure 1c, d demonstrates excess-entropy scaling for the diffusion coefficient in the 4:1 KA and 3:1 WS mixtures for three different densities and several temperatures; Supplementary Fig. S1 provides the corresponding figures for the viscosity. The 4:1 KA mixture is more commonly studied in the literature than

the 2:1 KA mixture, and the former model is therefore used to illustrate excess-entropy scaling. We focus here on the large A-particle diffusion coefficient; results for the B-particle diffusion coefficient are given in Supplementary Fig. S2. Consistent with previous studies[21,34,69], an excellent collapse with excess entropy is found, extending here to much lower diffusion coefficients than previously studied. Hereafter we focus on showing results for a fixed density only for each system.

**Composition excess-entropy scaling**. Excess-entropy scaling for a fixed composition was demonstrated in the previous section. However, composition is an extra variable besides density and temperature in the phase diagram of mixtures, and the question is therefore whether excess-entropy scaling can absorb this extra variable and still collapse data to a univariate function of the excess entropy $S_{ex}$. As mentioned, in light of the results of Krekelberg et al.[22] and from the fact that mixtures have rich phase diagrams, one does not a priori expect any collapse for different compositions.

Figure 2 shows the reduced viscosity $\widetilde{\eta}$ as a function of the excess entropy for the KA mixture, the WS mixture, the GLJ mixture, and the CuZr mixture, each plotted for several compositions. An almost composition-independent curve is found for all mixtures for a dynamic range extending over eight orders of magnitude in viscosity. This result cannot in an obvious way be explained by the quasiuniversality of single-component atomic liquids or by appealing to high temperatures where binary mixtures are expected to behave approximately as single-component liquids.

**Quasiuniversal excess-entropy scaling**. We proceed to investigate excess-entropy scaling relationships by comparing different systems. Figure 3a shows the reduced viscosity as a function of the excess entropy for all mixtures and compositions. Figure 3b shows the reduced A-particle diffusion coefficient. For reference we have also included data for the single-component LJ (SCLJ) liquid. Additional data for SCLJ are given in ref. [75], demonstrating that the same trend continues into the gaseous region.

For all investigated mixtures and compositions, a quasiuniversal relationship is observed for both viscosity and diffusion coefficient using the excess entropy as the relevant variable. Some deviations are found for the most supercooled states, depending on the mixture, and thus the use of the term quasiuniversal is appropriate as opposed to the nearly universal relationship observed for different compositions in Fig. 2. We conclude that quasiuniversality applies also for binary mixtures, contrary to expectation and previous studies.

To put the magnitude of the observed deviations into perspective, Figure 3b provides as a reference excess-entropy scaling for an almost sphere-like dumbbell molecule (DB; see grey data points with data taken from ref. [26]). This model also has $R$ above 0.90 for all investigated state points. Significant deviations are observed at higher temperatures and no quasiuniversality can possibly be established in the deeply supercooled region, indicating that the deviations between the different binary mixtures are relatively small. The departure from universality in the supercooled region is studied more closely below where it is found to correlate with the value of the density-scaling exponent.

How do the above quasiuniversality observations relate to those of Flenner et al.? Flenner et al.[68] observed a quasiuniversal breakdown of SE for five different binary atomic mixtures by scaling the relaxation time (a proxy for the viscosity) and the diffusion coefficient and plotting the diffusion coefficient against the relaxation time[68]. The SE relation in its traditional form is

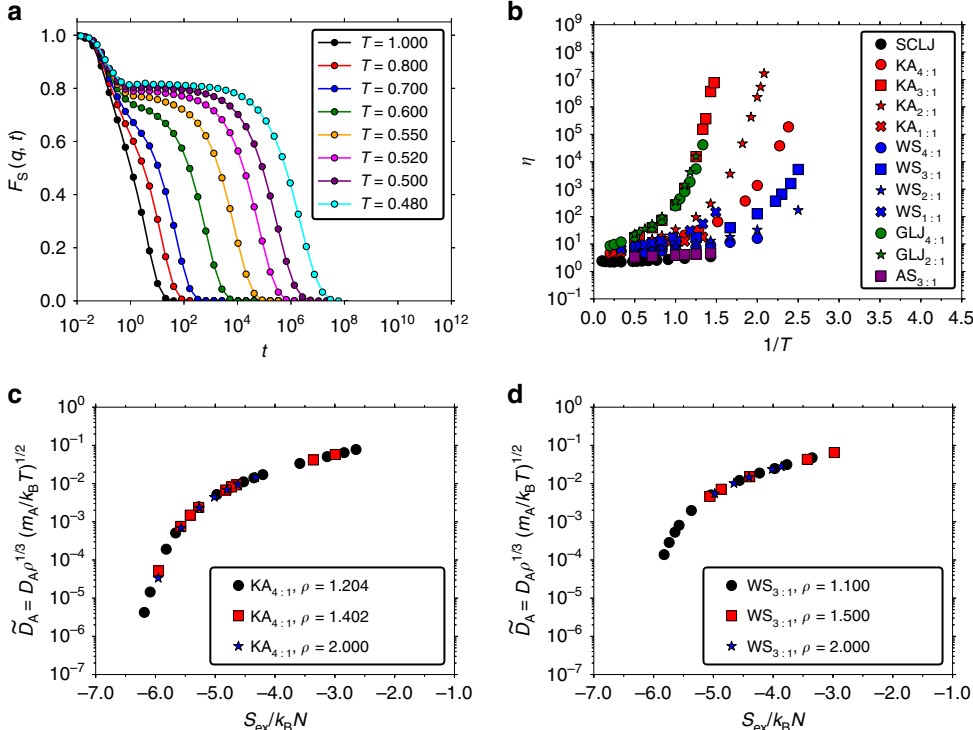

**Fig. 1 Supercooled dynamics and excess-entropy scaling. a** Temperature dependence of the self-part of the ISF for the 2:1 KA mixture ($\rho = 1.400$). The mode-coupling temperature is $T_{MCT} = 0.55$. **b** The viscosity $\eta$ as a function of $1/T$ for all mixtures. Significant non-Arrhenius temperature dependence is observed for all systems. The KAEXP and CuZr mixtures are omitted due to their different temperature scales (they show a similar behavior). The studied densities are given in "Methods". **c** Excess-entropy scaling for diffusion coefficient for the 4:1 KA mixture at the densities: $\rho = 1.204, 1.402, 2.000$. **d** Excess-entropy scaling for diffusion coefficient for the 3:1 WS mixture at the densities: $\rho = 1.100, 1.500, 2.000$. An excellent collapse is found for both systems, consistent with the isomorph-theory predictions[44].

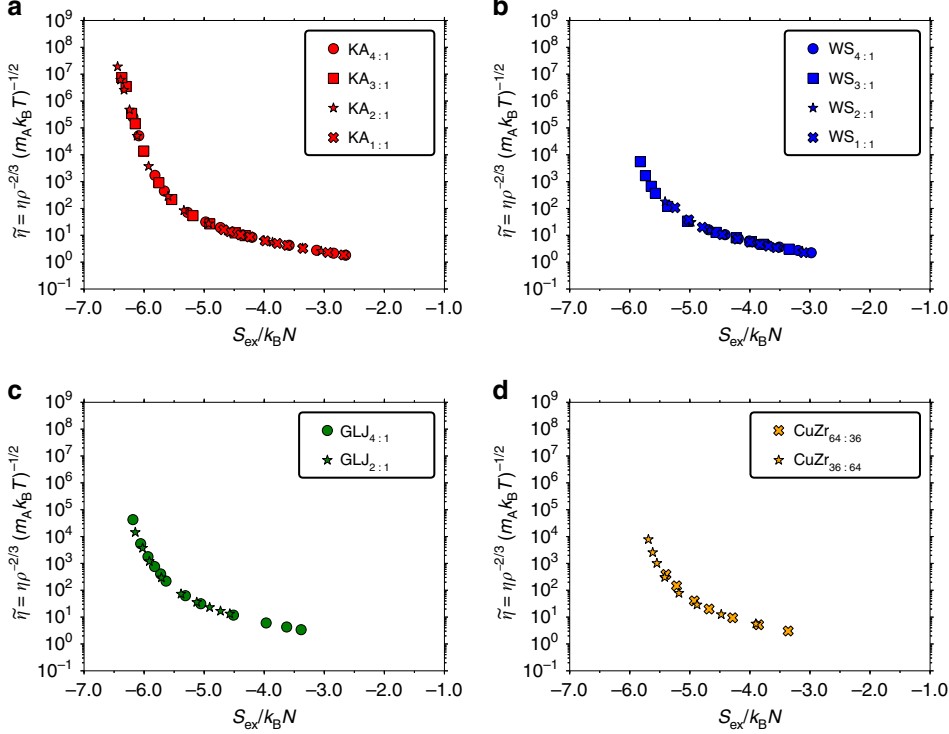

**Fig. 2 Reduced viscosity $\widetilde{\eta}$ as a function of $S_{ex}$ for several mixtures and compositions.** A universal curve is observed for each system with excess entropy. **a** KA mixtures. $\rho = 1.204$ for 4:1, $\rho = 1.400$ for 3:1, $\rho = 1.400$ for 2:1, $\rho = 1.450$ for 1:1. **b** WS mixtures. $\rho = 1.000$ for 4:1, $\rho = 1.100$ for 3:1, $\rho = 1.100$ for 2:1, $\rho = 1.296$ for 1:1. **c** GLJ mixtures. $\rho = 1.200$ for 4:1, $\rho = 1.350$ for 2:1. **d** CuZr mixtures. $\rho = 0.08$ Å$^{-3}$ for both compositions.

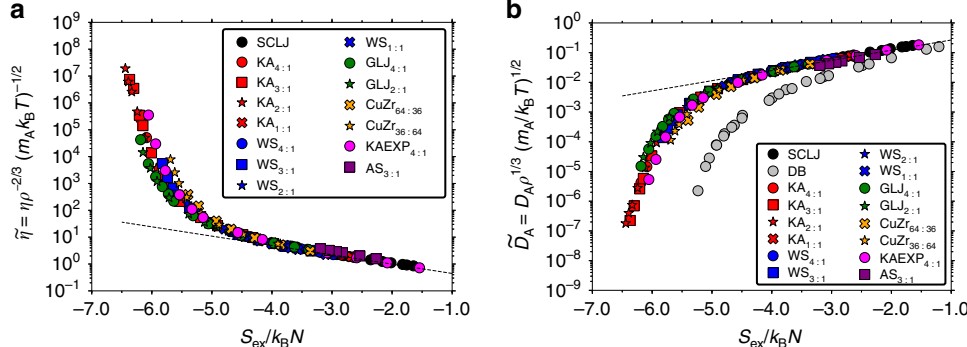

**Fig. 3 Reduced viscosity and A-particle diffusion coefficient as a function of $S_{ex}$ for all mixtures and compositions.** A quasiuniversal curve is observed in both cases with excess entropy. The black dashed lines give Rosenfeld single-component quasiuniversality Eq. (1). **a** Reduced viscosity $\widetilde{\eta}$ as a function of $S_{ex}$. **b** Reduced A-particle diffusion coefficient $\widetilde{D}_A$ as a function of $S_{ex}$. The grey data points (DB, data taken from ref. [26]) give reference data for an almost sphere-like dumbbell molecule that is easily supercooled; in this case no quasiuniversality is found. In comparison, the observed deviations between the different binary mixture data are relatively small.

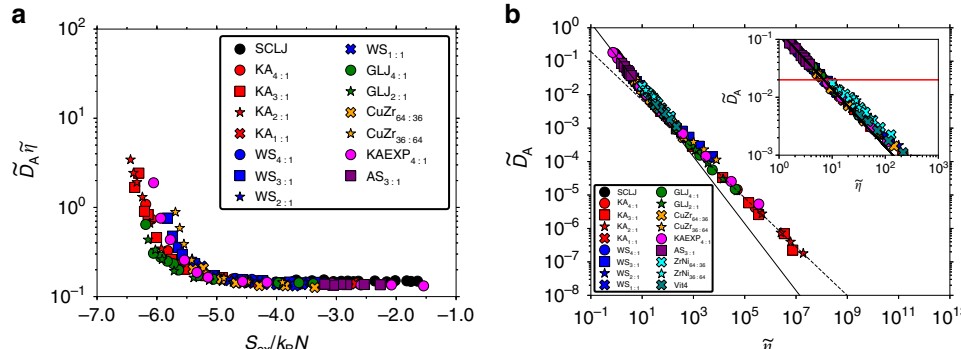

**Fig. 4 Breakdown of the reduced-unit SE relation $\widetilde{D}_A\widetilde{\eta} = 1/c\pi$. a** The product $\widetilde{D}_A\widetilde{\eta}$ as a function of the excess entropy $S_{ex}$ for all simulated systems. A quasiuniversal breakdown begins around $S_{ex}/k_B N \approx -5.0$. **b** $\widetilde{D}_A$ as a function of $\widetilde{\eta}$. The full black line is the SE relation with slope $-1$ fitted to 1:1 KA data, and the black dashed line is fractional SE with $\xi \approx 0.73$ fitted to deeply supercooled 2:1 KA data. The aquamarine crosses are experimental data for the binary metallic alloy $Zr_{64}Ni_{36}$ taken from Brillo et al.[83] and stars are from ref. [85] for $Zr_{36}Ni_{64}$; the diffusion coefficient of Ni was measured. The teal-colored data points (ref. [86]; see also text) are for the Vit4 ($Zr_{46.8}Ti_{8.2}Cu_{7.5}Ni_{10}Be_{27.5}$) five-component metallic glass former probing the diffusion of Ni/Ti/Cu. The inset zooms in on the SE breakdown with the horizontal red line indicating the approximate onset of SE breakdown. A quasiuniversal breakdown is noted around $\widetilde{D}_A \approx 2 \times 10^{-2}$. A similar conclusion is reached for the B-particles in Supplementary Fig. S2.

given by

$$D\eta = \frac{k_B T}{c\pi\sigma_H}, \quad (2)$$

in which $\sigma_H$ is the hydrodynamic diameter and $c$ is a constant. Assuming that the hydrodynamic diameter is not a constant but proportional to $\rho^{-1/3}$, the SE relation in reduced units becomes[76]

$$\widetilde{D}\widetilde{\eta} = \frac{1}{c\pi}. \quad (3)$$

A hydrodynamic diameter proportional to $\rho^{-1/3}$ was proposed by Zwanzig[77] and recently shown to be a consequence of the isomorph theory in the sense that Eq. (2) with a constant hydrodynamic diameter is inconsistent with isomorph theory while Eq. (3) is not[76]. We therefore focus on this expression for SE. For atomic mixtures, a possible generalization of the SE relation is to use the individual diffusion coefficients in Eq. (3), e.g., for the A-particle the diffusion coefficient $D_A$ is used[33] while the constant $c$ is expected to depend on the particle type.

The SE relation is now investigated for all the binary mixtures. Figure 3 documents a quasiuniversal relation for both $\widetilde{D}_A$ and $\widetilde{\eta}$ as a function of the excess entropy $S_{ex}$. This result implies that the product is also a quasiuniversal function of $S_{ex}$. Figure 4a shows

$\widetilde{D}_A\widetilde{\eta}$ as a function of the excess entropy for all investigated systems. We find a quasiuniversal breakdown of SE (i.e., departure from a constant value) around $S_{ex}/k_B N \approx -5.0$. A breakdown of SE has been correlated to the crossing of the so-called two-particle excess entropy and the excess entropy with temperature[69]. It would be interesting to check whether this observation holds for the systems studied here.

Figure 4b shows $\widetilde{D}_A$ versus $\widetilde{\eta}$ in a plot where the SE relation is a straight line with slope $-1$ (the full black line). Around $\widetilde{D}_A \approx 2 \times 10^{-2}$ the SE relation begins to break down for all systems and compositions. These data suggest that the relevant variable is the excess entropy which in a quasiuniversal way correlates to both viscosity and diffusion coefficient and hence also their product, defining the SE relation. Although the departure from universality for viscosity and diffusion coefficient go in opposite directions in Fig. 3, a specific value of $S_{ex}$ corresponds to a specific value of $\widetilde{D}_A$ or $\widetilde{\eta}$ due to the separate quasiuniversality of these two quantities. The breakdown is therefore bound to occur at more or less the same value of $\widetilde{D}_A$ or $\widetilde{\eta}$ for all the studied systems. Supplementary Fig. S2 provides the same figure for the B-particle diffusion coefficient in which case the same conclusion is reached. We conclude that quasiuniversality for binary mixtures can rationalize the observations of Flenner et al.[68] that SE breaks down at the same scaled relaxation time.

The excess entropy approach detailed here does not clarify the origin of the SE breakdown, other than it should occur in a quasiuniversal manner. Other theoretical approaches, such as dynamical facilitation or the random first-order transition theory, have the SE breakdown as a consequence of dynamical heterogeneity[15,64,65]. These theories provide predictions for the fractional SE exponent observed in Fig. 4b (see the black dashed line) which the excess entropy approach does not provide. We find that the fractional SE exponent for our most supercooled 2:1 KA data is $\xi \approx 0.73$, which interestingly is also the number found in simulations of the one-dimensional East model in dynamical facilitation[64]. Similar fractional SE exponents have been noticed before, but in this study we go almost four decades below $T_{MCT}$ and find an excellent agreement with the quasiuniversal excess-entropy scaling.

**Excess-entropy scaling in experiments.** A recent experimental study by Blodgett et al.[78] proposed an interesting universality for metallic liquids by scaling viscosity with the high-temperature limit $\eta_0$ and temperature with the onset of cooperative motion $T_A$. A good collapse of many different alloys was obtained in the high-temperature limit and close to the glass transition, motivated by avoided critical point theory (KKZNT)[5]. The authors therefore found to a good approximation $\eta/\eta_0 = F(T/T_A)$. For the alloys studied the authors noted on average that $\eta_0 \propto \rho$ and $T_A/T_1 \approx 1.075$, where $T_1$ is the liquidus (freezing) temperature. Recall that for binary mixtures the liquidus temperature specifies the temperature at constant pressure above which the system is completely liquid (the opposite being the so-called solidus temperature). For R-simple liquids, the temperature is given by $T = h(\rho)f(S_{ex})$ in which $h(\rho)$ is a function of density[79]. The freezing line is an approximate isomorph[44,80], and since an isomorph is characterized by $h(\rho)/T = $ const., one has $h(\rho) \propto T_f(\rho)$ with the reference isomorph being the freezing line[81,82]. The quasiuniversality found here explains the quasiuniversality found for metallic alloys since $T/T_A \approx T/T_1 \approx T/T_f(\rho) = f(S_{ex})$.

In Fig. 4b we plot quasielastic neutron scattering measurements of the Ni diffusion coefficient against the reduced viscosity for the binary metallic alloy $Zr_{64}Ni_{36}$ using data of Brillo et al.[83] (see also, e.g., ref. [84]) and similar data for $Zr_{36}Ni_{64}$ from ref. [85]. The reduced diffusion coefficients and viscosities for both $Zr_{64}Ni_{36}$ and $Zr_{36}Ni_{64}$ collapse nicely onto the quasiuniversal curve, reflecting the underlying quasiuniversal excess-entropy scaling relationship. The same figure also shows data for the Vit4 ($Zr_{46.8}Ti_{8.2}Cu_{7.5}Ni_{10}Be_{27.5}$) five-component metallic glass former from Yang et al.[86]. The Vit4 glass former also collapses nicely onto the quasiuniversal curve. This shows that quasiuniversality extends beyond the binary mixtures of main focus here. We return to this observation in the "Discussion".

For testing quasiuniversal excess-entropy scaling in experiments as in Fig. 3, the two-body entropy[32,87] could be used as a proxy, but a high-temperature study indicates that it is not always a good approximation[33]. For our data the two-body entropy is a somewhat worse correlator than the excess entropy and also weakens the correlation to the density-scaling exponent (see later section). We therefore emphasize that the scaling is correlated to the full excess entropy which is more difficult to calculate, unfortunately. Figure 4b provides an alternative procedure for testing quasiuniversality in experiments which avoids having to evaluate $S_{ex}$ explicitly.

**Additional tests for quasiuniversal behavior.** Rosenfeld quasiuniversality for single-component atomic liquids can be explained by appealing to the EXP pair potential, in terms of which other pair potentials under certain conditions may be

expanded[59]. For single-component systems quasiuniversality therefore implies not only quasiuniversal Rosenfeld scaling, but also Young and Andersen's structure-dynamics scaling principle[88,89], quasiuniversal freezing rules[90], invariance of the reduced viscosity along the melting line[91], and more. The single-component arguments do not, however, readily generalize to mixtures. In view of this, we proceed to test to which extent quasiuniversality holds for binary mixtures by checking whether the structure is similar amongst state points with similar dynamics, i.e., whether Young and Andersen's scaling principle applies.

Figures 5a, b compares two different compositions (4:1 and 2:1) for the KA mixture at state points for which the excess entropy and reduced diffusion coefficients are almost identical. For these state points there is less than 9% difference in reduced diffusion coefficient and <0.5% difference in excess entropy. Nevertheless, we find that the AA-particle radial distribution functions (RDFs) show rather large deviations between these two systems, certainly much larger than what is normally found for single-component atomic systems[88,89]. Even larger deviations are found for the AB and BB-particle RDFs in Supplementary Fig. S3.

This observation implies that two-body correlations do not uniquely determine the supercooled dynamics and thus that many-body correlations are important for the dynamics of the system[10,92,93]. The rather large difference in RDFs between the two compositions might also be anticipated from the relevance of the locally favored structures (bicapped square antiprisms) for the dynamics in these mixtures[94]. Furthermore, this anticipation is supported by a connection between decoupling of component dynamics, dynamical heterogeneity, and development of different local medium-range-like ordering in the supercooled regime for certain alloys where the local ordering is directly detectable in the RDFs[95].

Figure 5c, d compares AA-particle RDFs and MSDs amongst the KA and WS mixtures at the same 3:1 composition. The state points have <1% difference in reduced diffusion coefficient and <0.3% difference in excess entropy. We find also here rather large variations of the AA-particle RDFs and even larger ones for the AB and BB-particle RDFs (Fig. 5e, f).

Supplementary Fig. S4 compares the distribution of Voronoi volumes in the liquid for the same systems and state points as above, showing also here clear differences. The quasiuniversality found in supercooled binary mixtures thus appears to be more subtle than the quasiuniversality observed in single-component atomic liquids at high temperatures. Future work should focus on clarifying the nature behind this observation in the supercooled regime which could be related to local orderings in the supercooled liquid[95].

**Departure from universality.** Figure 3 displayed some deviations from universality in the scaling in the supercooled regime. This section considers these deviations in more detail. Figure 6 shows the reduced viscosity and diffusion coefficient as a function of the excess entropy, where each data point is colored after its value for the density-scaling exponent[44].

$$\gamma = \frac{\langle \Delta U \Delta W \rangle}{\langle (\Delta U)^2 \rangle}. \tag{4}$$

The departure from universality in the supercooled regime correlates with the value of the density-scaling exponent, with a smaller value of $\gamma$ moving the curve up for viscosity and down for diffusion, the opposite being the case for larger $\gamma$-values. More similar $\gamma$-values—irrespective of mixing rules, interaction types, etc.—therefore conform to a more universal scaling in the

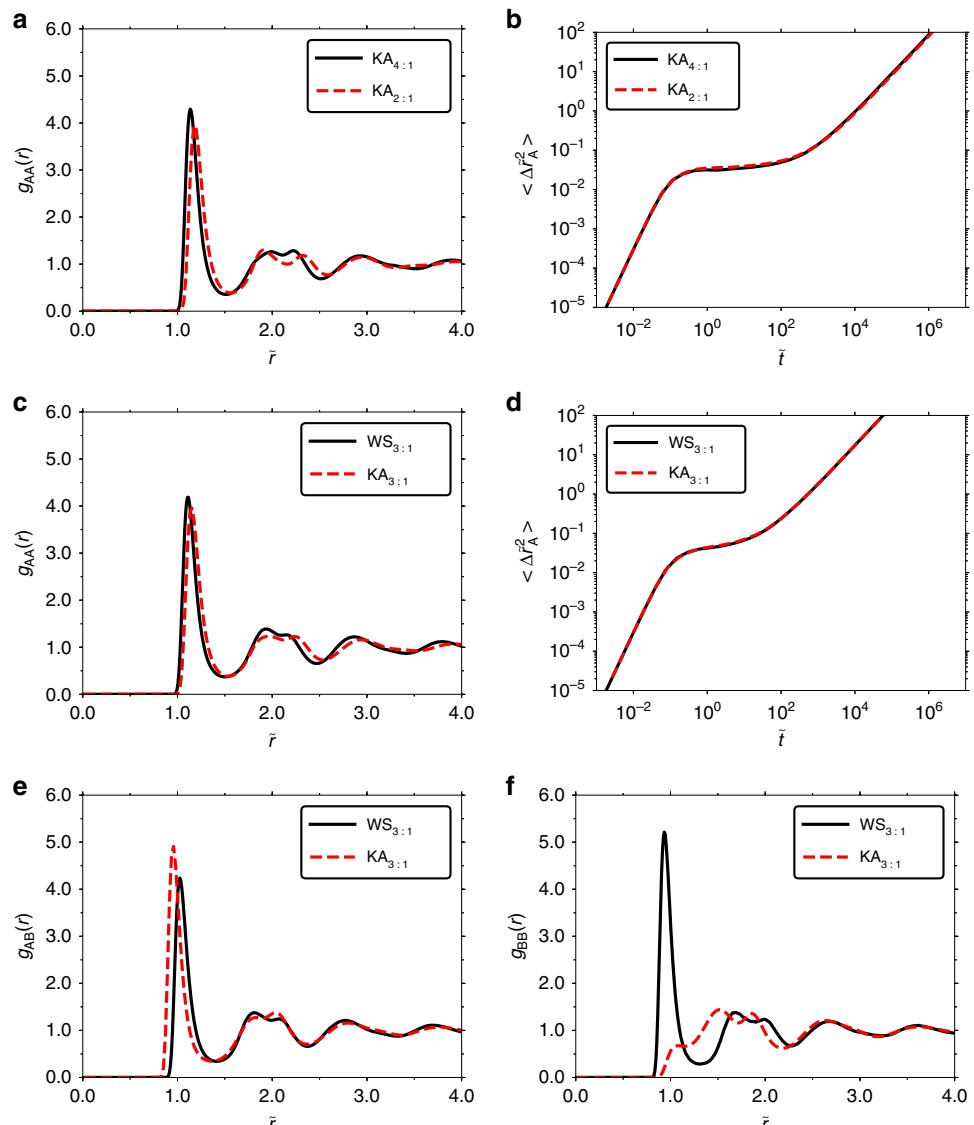

**Fig. 5 Tests of radial distribution function (RDF) quasiuniversality where $\widetilde{r} = \rho^{1/3}r$ and $\widetilde{t} = t\rho^{1/3}(k_B T/m_A)^{1/2}$. a** AA-particle RDFs for the KA mixture for two different compositions (4:1 and 2:1) at state points with almost identical $S_{ex}$ and $\widetilde{D}_A$. For 4:1 KA: $\rho = 1.204$, and $T = 0.440$ and for 2:1 KA: $\rho = 1.400$, and $T = 0.550$. **b** A-particle MSDs with details given in **a**, demonstrating almost identical reduced diffusion coefficients. **c** AA-particle RDFs for the KA and WS mixtures at the same 3:1 composition for state points with almost identical $S_{ex}$ and $\widetilde{D}_A$. For 3:1 KA: $\rho = 1.400$ and $T = 0.900$ and for 3:1 WS: $\rho = 1.100$ and $T = 0.415$. **d** A-particle MSDs with details given in **c**. **e** AB-particle RDFs with details given in **c**. **f** BB-particle RDFs with details given in **c**.

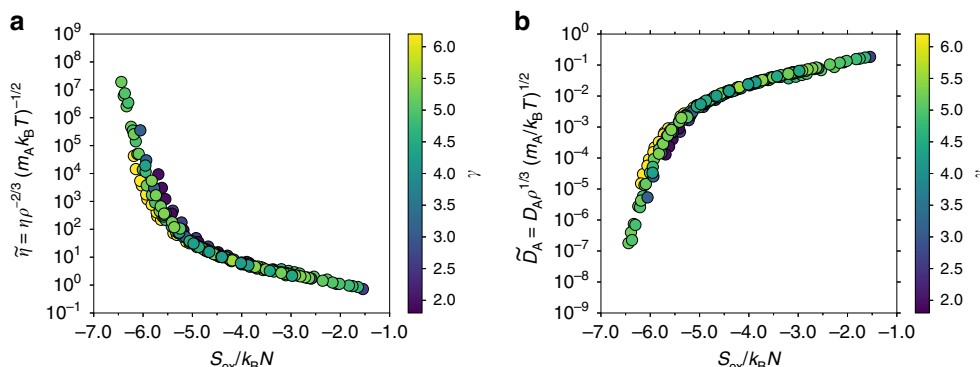

**Fig. 6 Relation between the density-scaling exponent and departure from universal excess-entropy scaling.** The color coding represents the value of the density-scaling exponent $\gamma$, Eq. (4). **a** $\widetilde{\eta}$ as a function of $S_{ex}/k_B N$. **b** $\widetilde{D}_A$ as a function of $S_{ex}/k_B N$.

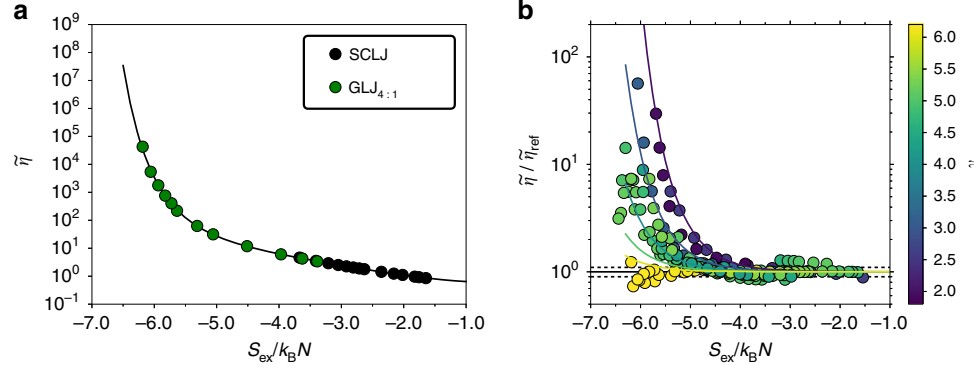

**Fig. 7 Accounting for the departure from universality. a** Reference curve $\widetilde{\eta}_{ref}$ given by Eq. (5) based upon viscosity data for the SCLJ and 4:1 GLJ systems. **b** Ratio of viscosity data for all binary mixtures to that obtained from the reference curve, colored by the value of $\gamma$. The dashed lines indicate the values 0.9 and 1.1, and the colored lines are the isolines $\gamma = (2, 3, 4, 5, 6)$ [see Eq. (7) in text].

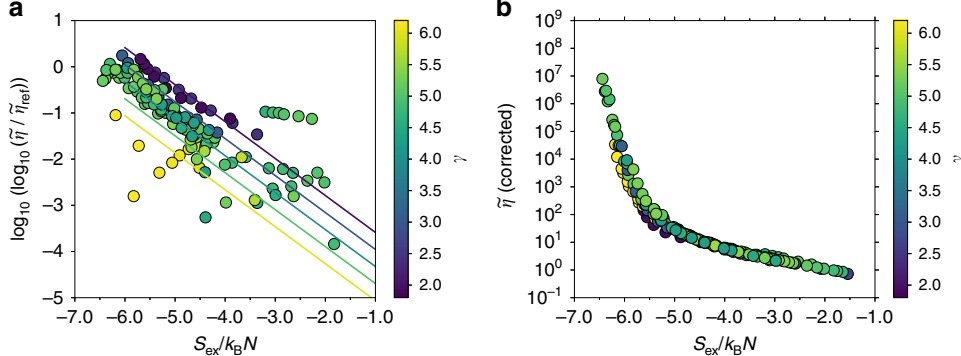

**Fig. 8 Correcting for the deviation from universality. a** The correction plotted as a function of $S_{ex}/k_BN$, where the colored lines are isolines of $\gamma = (2, 3, 4, 5, 6)$. Shown are data with $\widetilde{\eta} > \widetilde{\eta}_{ref}$. **b** The corrected viscosities using Eq. (7) as a function of $S_{ex}/k_BN$. A better collapse is obtained compared to Fig. 6a.

supercooled regime. Based on this observation, an empirical correction using only knowledge of $\gamma$ can be formulated.

In order to investigate the departure from universality more closely, we use an empirical reference-curve fit to the viscosity data for the SCLJ and 4:1 GLJ systems representing the leftmost part of the data set. The following functional form is used

$$\log_{10}(\log_{10}(\widetilde{\eta}_{ref} + 1)) = \sum_{i=0}^{4} c_i (S_{ex}/k_BN)^i, \quad (5)$$

with the best-fit coefficients $c_0 = -0.601 \pm 0.0541$, $c_1 = 0.267 \pm 0.0657$, $c_2 = 0.256 \pm 0.028$, $c_3 = 0.0568 \pm 0.005$, and $c_4 = 0.00448 \pm 0.000319$, where the number after the $\pm$ indicates the estimated standard deviation. A plot of the viscosities for the SCLJ and 4:1 GLJ systems along with the reference curve is shown in Fig. 7a. Figure 7b displays for all binary mixtures the ratio of the viscosity to that obtained from the reference-curve fit $\widetilde{\eta}_{ref}$ of Eq. (5). There is clearly a systematic trend in $\gamma$, though a few exceptions can also be found.

The excess-entropy dependence of the viscosity is super-Arrhenius. A pragmatic approach for linearizing the data is therefore to consider $\log_{10}(\log_{10}(\widetilde{\eta}/\widetilde{\eta}_{ref}))$; these values are shown in Fig. 8a. The slope in these coordinates, for a given value of $\gamma$, is approximately constant with a value of $-0.8$. The intercept value $b(\gamma)$ was found to be acceptably modeled by linear interpolation between the intercept values for $\gamma_{min} = 1.9$ and $\gamma_{max} = 6.1$

$$b(\gamma) = -0.369\gamma - 3.649, \quad (6)$$

yielding the overall correction of

$$\log_{10}\left(\log_{10}\left(\frac{\widetilde{\eta}}{\widetilde{\eta}_{ref}}\right)\right) = -0.8(S_{ex}/k_BN) + b(\gamma). \quad (7)$$

Figure 8b shows the corrected data for viscosity using this three-parameter expression; to apply the correction only knowledge of $\gamma$ is needed. A better collapse is obtained compared to Fig. 6a.

## Discussion

For both the diffusion coefficient and the viscosity, the current study has detailed an almost composition-independent relation to the excess entropy for a given system, as well as a quasiuniversal relation amongst different systems. As the viscosity and diffusion coefficient both show quasiuniversality, their product is also quasiuniversal. The SE relation for viscosity and diffusion coefficient must then break down at the same reduced relaxation time or, equivalently, the same value of the excess entropy. Our observations therefore rationalize the universal SE breakdown results of Flenner et al.[68]. The departure from universality correlates with the density-scaling exponent $\gamma$ with more similar $\gamma$-values exhibiting a more similar scaling. This may provide a hint towards explaining the observed deviations in the future.

The isomorph theory states that certain quantities in reduced units are invariant along constant excess-entropy curves in the thermodynamic phase diagram. This fact leads immediately to excess-entropy scaling as described in the Introduction. Does this necessarily imply a causal link between the excess entropy and transport coefficients? The answer is no because one can in

principle take the opposite view and posit that transport coefficients control the excess entropy.

Quasiuniversality is often explained by referring to the HS model[96]. The HS model was recently questioned as a good reference system as this model cannot account for all quasiuniversality observations[58–60,96]. Likewise, we do not believe that the HS model can explain our observations, even by introducing two different spheres, as we considered both very soft and very harsh repulsive pair potentials, highly nonadditive and exothermic mixtures, and mixtures with effective medium interactions. It is also not obvious how the EXP pair-potential arguments for single-component atomic liquid's quasiuniversality can be extended to explain our observations. The fact that the RDFs and Voronoi volumes were observed not to be the same for state points with very similar dynamics and excess entropy, points to a possibly more complex kind of quasiuniversality than that of single-component systems.

An open question is why quasiuniversality is only observed in atomic mixtures but not also in, e.g., single-component molecular systems, even for small molecules[25]. A conjecture is that by removing certain degrees of freedom (e.g., vibrational degrees of freedom) one might be able to unravel quasiuniversality in molecular systems[52,97]. Another relevant question is how a large mass or size ratio would influence the scalings. We studied up to a factor of two in mass ratio and up to a factor of three in size ratio between the constituent particles. A recent study[98] has shown that both cases can have a nontrivial effect on the dynamics of supercooled liquids. Binary mixtures with very large size ratios are not expected to be R-simple[53]. A possible explanation for the lack of scaling in some of the results of Krekelberg et al.[22] is then that these systems are not R-simple.

A limitation of the current study is the focus on binary atomic mixtures. Figure 4b included data for the Vit4 ($Zr_{46.8}Ti_{8.2}Cu_{7.5}Ni_{10}Be_{27.5}$) five-component glass former[86] and showed a good collapse for the diffusion coefficient of Ni/Ti/Cu. We therefore anticipate that mixtures with several components are also covered by the current quasiuniversality relation discovered for binary mixtures. However, it has been observed in some metallic glass formers that SE can apply for one specific component but not for others (see, e.g., ref. [99]). This behavior could be related to the development of different local orderings in the liquid as seen for a Cr-based alloy[95]. Fundamental questions are therefore: For which component does quasiuniversality hold—and why?

Related to this topic, a recent study found similar structure and dynamics for weakly polydisperse systems sharing the same repulsion when compared at the same $T/T_g$ value[100], where $T_g$ is the glass transition temperature. These results are consistent with our conjecture that quasiuniversality extends beyond binary mixtures. Due to the extremely time-consuming simulations of this paper, this intriguing topic is left for future research.

A long-standing issue in the study of supercooled liquids is what controls the dynamics. We find in this study that the excess entropy correlates well with the viscosity and diffusion coefficient for a wide range of binary mixtures, including metallic alloys. Furthermore, evidence has been presented that these results may extend beyond binary mixtures. The novel multicomponent metallic alloys being designed today cannot be comprehensively studied in experiments because of the immense number of possible mixture compositions[101,102]. The approach proposed in this paper offers a means of providing predictive guidance for the transport properties of novel alloys since the quasiuniversal excess-entropy scaling is expected to hold for these liquids. As a result, it is a realistic hope that excess-entropy scaling may facilitate the design of future metallic glasses.

## Methods

**Simulation details**. Molecular dynamics computer simulations were carried out using Nvidia Geforce GTX 1080 graphics cards and the Roskilde University Molecular Dynamics (RUMD) package, version 3.4, in single precision[71]. Very long equilibration runs (the longest ones lasting more than 12 months) were used to ensure equilibrium before initiating production runs. The equilibrium and production runs were in the NVT ensemble with Nosé–Hoover thermostatting[103]. Possible crystallization was checked using various order parameters, potential energy, etc. It was confirmed after equilibration that the results are reproducible by running the production-run simulations at least twice.

**Binary mixtures**. We studied six different binary mixtures: the Kob–Andersen binary Lennard–Jones (KA) mixture, the Wahnström (WS) mixture, the GLJ mixture, the KA exponential pair potential (KAEXP) mixture, alloys of copper and zirconium (CuZr), and a size asymmetric (AS) mixture. One or several compositions were studied for each mixture. We focused mainly on one density and varied the temperature, but for the 4:1 KA and 3:1 WS mixtures density was additionally varied. For reference the SCLJ liquid was also simulated ($\rho = 0.850$ and $N = 1024$). All pair potentials used a shifted-potential cutoff, except for KAEXP which used a shifted-force cutoff[47,60,104,105].

The KA mixture[106,107] uses the LJ pair potential $v_{\alpha\beta}(r) = 4\epsilon_{\alpha\beta}[(\sigma_{\alpha\beta}/r)^{12} - (\sigma_{\alpha\beta}/r)^6]$ with parameters: $\sigma_{AA} = 1$, $\sigma_{BB} = 0.88$, $\sigma_{AB} = 0.80$, and $\epsilon_{AA} = 1$, $\epsilon_{BB} = 0.50$, $\epsilon_{AB} = 1.50$. The mass is unity for both particles and the cutoff is $r_{cut} = 2.50\sigma_{\alpha\beta}$. The density of interest was for 4:1 KA: $\rho = 1.204$, 3:1 KA: $\rho = 1.400$, 2:1 KA: $\rho = 1.400$, 1:1 KA: $\rho = 1.450$. The density was changed to avoid negative pressure upon cooling, and we note the composition for 4:1 KA is very slightly nonstandard with 4.019:1. The particle numbers were $N = 1024$, 10000, 10002, 10000 for 4:1, 3:1, 2:1, 1:1 KA, respectively. The time step was in the range $\Delta t = 0.001–0.0035$, depending on temperature. The longest production runs were 69 billion time steps.

The WS mixture[108] uses the same pair potential and cutoff as the KA mixture but applies the Lorentz–Berthelot mixing rules with $\sigma_{AA} = 1.0$, $\sigma_{BB} = 0.833$, and $\epsilon_{AA} = \epsilon_{BB} = \epsilon_{AB} = 1$. The masses are $m_A = 2.0$ and $m_B = 1.0$. The density of interest was for 4:1 WS: $\rho = 1.000$, 3:1 WS: $\rho = 1.100$, 2:1 WS: $\rho = 1.100$, 1:1 WS: $\rho = 1.296$. The particle numbers were $N = 1024$, 1000, 1002, 1024 for 4:1, 3:1, 2:1, 1:1 WS, respectively. The time step was in the range $\Delta t = 0.001–0.0025$. The longest production runs were 268 million time steps.

The GLJ mixture varies the exponents of the LJ pair potential but keeps the location of the minimum fixed; in our case $m = 12$ and $n = 10$, where $m$ and $n$ are the repulsive and attractive exponents of the GLJ pair potential, respectively. The GLJ pair potential is given by $v_{\alpha\beta}(r) = \epsilon_{\alpha\beta}/(m - n)[n(\sigma_{\alpha\beta}/r)^m - m(\sigma_{\alpha\beta}/r)^n]$. The parameters are: $\sigma_{AA} = 2^{1/6}$, $\sigma_{BB} = 0.88 \times 2^{1/6}$, $\sigma_{AB} = 0.80 \times 2^{1/6}$, and $\epsilon_{AA} = 1$, $\epsilon_{BB} = 0.50$, $\epsilon_{AB} = 1.50$ with $r_{cut} = (2.50/2^{1/6})\sigma_{\alpha\beta}$. The density of interest was for 4:1 GLJ: $\rho = 1.200$, 2:1 GLJ: $\rho = 1.350$. The particle numbers were $N = 1000$, 1002 for 4:1, 2:1 GLJ, respectively. The time step was in the range $\Delta t = 0.001–0.0025$. The longest production runs were 1.1 billion time steps.

The KAEXP mixture uses the same parameters as the KA mixture but replaces the LJ pair potential with repulsive exponential pair potentials given by $v_{\alpha\beta}(r) = \epsilon_{\alpha\beta} \exp[-r/\sigma_{\alpha\beta}]$. The cutoff is $r_{cut} = 4.50\rho^{-1/3}$, i.e., the cutoff depends on density. The density was $\rho = 0.001$ for 4:1 composition. The particle number was $N = 1024$ and the time step was $\Delta\tilde{t} = 0.0025$ in (macroscopically) reduced units. The longest production runs were 4.3 billion time steps. The single-component EXP pair-potential liquid was studied in refs. [60,105].

The CuZr mixture was simulated using the Effective Medium Theory (EMT) for metallic alloys[109,110]. EMT is a semiempirical many-body potential derived from DFT that offers a significant advantage over, e.g., most standard Embedded Atom Method (EAM) potentials since EMT does not require a tabulated format for the potential. The unit system used for CuZr is with the length scale of angstrom Å, mass dimension of atomic mass unit u, and energy scale of electron volt eV. We studied the compositions $CuZr_{64:36}$ and $CuZr_{36:64}$ at the density $\rho = 0.08$ Å$^{-3}$, and the particle number was $N = 1000$ for both compositions. The time step was $\Delta t \approx 7.13$ fs. The longest production runs were 400 million time steps.

The AS mixture is governed by the LJ pair potential with $\sigma_{AA} = 1.00$, $\sigma_{AB} = 0.65$, $\sigma_{BB} = 0.30$. $\epsilon_{AA} = 1.00$, $\epsilon_{AB} = 1.40$, $\epsilon_{BB} = 0.80$. $m_A = 2.0$ and $m_B = 1.0$. $r_{cut} = 2.50\sigma_{\alpha\beta}$ and $\rho = 1.100$ at 3:1 composition. For this kind of size disparity (more than a factor of three) it is difficult to avoid crystallization by phase separation even with a negative heat of mixing. The particle number was $N = 1000$ and the time step was $\Delta t = 0.0025$. The longest production runs were 67 million time steps.

**Analysis**. The diffusion coefficients of each particle type were obtained from fitting their respective long-time mean-square displacements to the Einstein relation. The shear viscosities were obtained from integrating the shear-stress time autocorrelation function via the Green–Kubo relation

$$\eta = \frac{V}{k_B T} \int_0^\infty \langle S_{\alpha\beta}(0)S_{\alpha\beta}(t)\rangle dt, \qquad (8)$$

where $S_{\alpha\beta}$ is the $\alpha\beta$-component of the stress tensor ($\alpha \neq \beta = x, y, z$), $V$ is the volume, and $\langle \ldots \rangle$ denotes an ensemble average. All three off-diagonal stress tensor components were averaged for better statistics. The value of the viscosity was

extracted from the first maximum of the integral which corresponds to the plateau value obtained in the running integral of Eq. (8). The self-part of the ISF was evaluted from $F_s(\mathbf{q}, t) = \langle \exp[i\mathbf{q} \cdot \Delta \mathbf{r}_i] \rangle$, where $\mathbf{r}_i$ is the position of particle $i$, and $\mathbf{q}$ is the wave vector. The length of the wave vector is given by the position of the first peak of the static structure factor.

The excess entropy $S_{ex}$ was calculated from the thermodynamic relation

$$F_{ex} = U_{ex} - TS_{ex}, \qquad (9)$$

using thermodynamic integration, where $F_{ex}$ is the excess Helmholtz free energy, and $U_{ex} \equiv U$ is the potential energy. Application of thermodynamic integration to supercooled liquids is standard[6,34,38]. First, a path at a high temperature $T_{ref}$ above the critical point was chosen, integrating the equation

$$W = \left( \frac{\partial F_{ex}}{\partial \ln \rho} \right)_T, \qquad (10)$$

from low density (the ideal gas) to the density of interest $\rho$ in order to obtain $F_{ex}(\rho, T_{ref})$, in which $W$ is the virial defined by $W = PV - Nk_BT$. Both $U$ and $W$ were obtained from the actual simulations. Afterwards, a path at the constant density $\rho$ was simulated, integrating from $T_{ref}$ to $T$ to obtain $F_{ex}(\rho, T)$ using the identity

$$U_{ex} = \left( \frac{\partial (F_{ex}/T)}{\partial (1/T)} \right)_\rho. \qquad (11)$$

We confirmed that the results for $S_{ex}$, within half a percent, are independent of the thermodynamic path as well as of the applied discretization of density and temperature. Larger error bars on $S_{ex}$ were found for the CuZr mixtures than for the other systems due to a nonmonotonic behavior at very low densities.

## Data availability

The data in csv file format that support the findings of Figs. 1–4 and 6–8 are available as Supplementary information. Equilibrated starting configurations and RUMD scripts are available at https://doi.org/10.18434/M32244.

## Code availability

The RUMD package is available online at http://rumd.org/.

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

## Acknowledgements
T.S.I. and J.C.D. are supported by the VILLUM Foundation's *Matter* grant (No. 16515). We are indebted to Nick Bailey, Lorenzo Costigliola, Harold W. Hatch, David Heyes, Ken Kelton, Mohammed H. Mousazadeh, Thomas B. Schrøder, Thomas Voigtmann, and Fan Yang for many useful discussions and suggestions.

## Author contributions
T.S.I. designed research; I.H.B., and T.S.I. performed research; I.H.B., J.C.D., and T.S.I. analyzed data; and T.S.I. wrote the paper with input from J.C.D.

## Competing interests
The authors declare no competing interests.
