## [Peer Review File · Nature Communications]

REVIEWER COMMENTS

Reviewer #1 (Remarks to the Author):

The manuscript reports an extensive numerical study on excess entropy scaling of transport coefficients in supercooled liquid mixtures. The authors show that excess entropy scaling holds in mixtures across compositions and explain this in terms of the "R-simple" nature of the studied systems. Previous work (Ref.22) focused on mixtures of hard spheres or Gaussian particles, which are not R-simple and for which excess entropy scaling does not hold. Additionally, the authors provide some evidence of "quasi-universal" behavior across mixtures with different interactions, that is, diffusion and viscosity collapse on an approximate master curve $f(S_{ex})$, which is a function of the excess entropy S_{ex} alone. The analysis carries over to experimental data for multi-component metallic alloys.

The study is supported by a tremendous numerical effort, which significantly extends previous attempts to assess entropy scaling in liquid mixtures. The paper is also very well written and provides a clear and compact presentation of entropy scaling approaches to the dynamics of supercooled liquids. It is a valuable contribution to the field and may also serve as dive-in introduction for non-specialists. However, I have two main criticisms and a few minor points that I ask the authors to address before I can recommend publication.

1) My main problem is with the notion of "quasi-universality", which I find ambiguous. Several other theoretical approaches to the glass transition predict generic functional forms and scaling relations. Small deviations from collapse can be used to favor one approach over another (cf. Elmatad et al. JPCB 2010) and are interpreted as a failure of the theory. Given the sizable deviations seen in Fig. 3, why should we grant excess entropy scaling the privilege of holding "quasi-universally", as opposed to "failing"? I agree that entropy scaling has no free parameters, still it is not clear to me that these deviations are small enough to claim that the scaling holds "quasi-universally". The authors make no effort to assess whether the observed deviations are small or large in a quantitative sense, nor if these deviations can be accounted for by some generalized scaling.

Here is one suggestion to address this point. Suppose you take different mixtures based on inverse power interactions (truly R-simple): would entropy scaling hold exactly across them? If this is the case, one could try to rationalize deviations from collapse in Fig.3 as due to the degradation of pressure-energy correlations. Stated differently, liquids that are more R-simple would collapse more universally than the others. The local pressure-energy correlation coefficient would quantify these deviations. If this were true, one would have a well-defined reference behavior [providing the exact shape of $f(S_{ex})$] and a quantitative measure to account for deviations. From Fig.3 one recognizes for instance that deviations are most pronounced for CuZr, for which the pressure-energy correlation coefficient is lower than 90%.

2) There is an overall ambiguity as to what one really learns from excess entropy scaling. The authors start off in p.1 by saying "This theoretical framework does not in general, however, suggest any causal link among isomorph invariants". However, in the conclusions they end up saying "a long-standing issue in the study of supercooled liquids is what controls the dynamics. We find that the excess entropy not only controls the viscosity and the diffusion coefficient [...] but also controls them in a quasiuniversal way." Why should excess entropy, as opposed to any other isomorph invariant, "control" the dynamics? Perhaps it's a matter of language, but "controlling" the dynamics implies to me some causal link, which is not proven yet I think. Please clarify.

Minor points:

- p.3: when presenting the overview of the results for the 2:1 Kob-Andersen mixture, it would be good to remind the T_{MCT} value for this non-canonical composition.

- p.3 The transition between "Excess entropy scaling" and "Composition-independent excess entropy scaling" sections is a bit awkward. Composition-independent scaling is already shown to hold for diffusion (Fig.1c and 1d), but the section "Composition-independent excess entropy scaling" starts as this were a new result.

- It would be good to clarify the origin of the discrepancy with the findings of Krekelberg et al. in Ref. 22 (which focused on non R-simple mixtures).

- p.5: "A specific value of S_{ex} corresponds to a specific value of De due to quasiuniversality, and the breakdown is therefore bound to occur at more or less the same value of $D A$ or ηe for all systems." and also on p. 9 "As the viscosity and diffusion both show quasiuniversality, their product must also show the same quasiuniversality."

The authors are forcibly ignoring the role of fluctuations here. D and η show quasi-universality essentially by compensation of errors, since the /deviations/ from universality go in opposite directions (opposite signs) for D and η , see Fig. 3.

- p.5: "We thus conclude that quasiuniversality for binary mixtures explains the observations of Flenner et al. (also p.9)"

The main claim of Flenner et al. was that the features of dynamic heterogeneities are universal in the studied range of temperatures. This is a much stronger claim than "all systems deviate from SE around some S_{ex} value". Please rewrite to clarify the precise connection with Ref.65

- Related to point 1) above, it would be useful to report the U-W correlation coefficients in the big table in the SI. Also, it would be useful to provide that table in a format suitable for further data analysis, like in csv format.

- The authors mention that the longest simulations lasted more than 12 months on GPU. Given the single precision arithmetics used in the force calculation, how could the stability of the simulations be enforced (and how was it assessed) over such a long time?!

--

Daniele Coslovich

April 12, 2020

Report on

Quasiuniversal Excess Entropy Scaling in Supercooled Binary Mixtures

Ian H. Bell, Jeppe C. Dyre, and Trond S. Ingebrigtsen

The present work reports on investigations of the quasiuniversality of transport properties versus excess entropy for a large class of binary mixtures. From the pioneering work of Rosenfeld who first highlight the possibility of a universal excess entropy scaling of transport properties such as diffusion, shear viscosity, and thermal diffusivity, a wealth of subsequent studies were devoted to testing such a law in various situations and systems. Nevertheless, during the three decade history of this still open issue, well described by the authors, very little efforts have been given to the investigation of the possible extension of this scaling law in deep undercooling conditions. And this is precisely the aim of the present work. For this purpose, numerically intensive molecular dynamic simulations for various binary mixtures and several compositions of them were performed in order to cover up to eight orders of magnitude variation of the viscosity for instance in the undercooled region. The authors found a quasiuniversal excess entropy scaling independent of composition and nature of the interactions. Such a finding seems to be correlated to the fact that they can be characterized as R -simple liquids with a correlation coefficient R larger than 0.9. A link with the Breakdown of the Stokes-Einstein relation is then proposed. The conclusions are clear and outcome and further suggestions of research lines are appealing. For all these reasons, I would recommend publication of this paper in Nature Communications. I Suggest the authors to consider the following remarks.

1. A significant place in the introduction is given to concept of isomorphism developed by the authors essentially in the last decade. Authors mention that the dimensionless excess entropy is invariant along isomorphs, and as so do the transport coefficients justifies theoretically *a priori* a universal scaling law *à la* Rosenfeld. However, this aspect is not really developed the paper. Probably it would be worth considering this aspect more deeply, as one can realize that the cooling process to reach deep undercooling states are probably not located on isomorphs.
2. One aspect of the present simulation data is the decoupling of the diffusion coefficients of the two species, as can be seen in the tables of the supplement materials. This aspect, in relation to the quasiuniversality, has not been developed in the paper. However it seem that this decoupling is related to the local structural and chemical ordering which can be different around each species (see for instance: npj Computational Materials **3**, 33, 2017, and related works). The differences in radial-distribution functions shown in Fig. 5 could be a manifestation of that, and might have a more direct effect on the dynamics, and also on the breakdown of the Stokes-Einstein relation. This may deserves a comment and considering both species for the diffusion.

Reviewer #3 (Remarks to the Author):

This manuscript reports computer simulations seeking to demonstrate the emergence and validity of a quasi-universal scaling relation (based on excess entropy) in supercooled binary (simple) mixtures.

This work is very fundamental and touches on one of the long-standing open questions in physical chemistry, i.e. the nature of the glass transition. While it could be argued that the practical relevance of this research is not immediately clear, these sort of advances bear a substantial impact across different fields - I would thus judge this work of broad enough interest to be published in Nature Communications.

To strengthen this point, I would note that the paper is very well written: the introduction is very clear, approaching the problem in a very logical way while referencing most of the relevant literature. The results are presented in a simple and logical way, with no embellishments while making for a truly enjoyable reading.

Onto the technical side of things, the results are robust and the authors took special care in cross-validating them with the existing literature data. Given the diverse array of systems they have investigated, the quasi-universality of the scaling relation they put forward is hard to dispute.

I do have a few issues with the current version of the manuscript, though, namely:

- The "quasi-universality" labelling is indeed adequate, given the deviations of ~ 2 orders of magnitude observed in e.g. Fig. 3a at strong supercooling. However the authors failed to provide any explanation whatsoever as to what could be the origin of said deviations: this is a crucial point that deserves further investigation. For instance, the authors could think of potential avenues to exacerbate these deviations - perhaps the R coefficient is playing a role? I feel this is an important piece of the puzzle that truly cripples the impact of this work.
- Using the radial distribution function alone to probe the extent of the validity of the Young-Andersen principle is not enough. At the very least the authors should offer the customary complementary analysis of the distribution of the volumes of the Voronoi polyhedra within the liquids, as the radial distribution function does not uniquely characterise a disordered system. I also believe there is a huge opportunity here to elaborate on the impact of potential structural features on the scaling law put forward by the authors, whom are in fact in a good position to make a connection with what they define as "the nature behind the SE breakdown". A connection with some of the theories dealing with the nature of the SE breakdown (Dynamical Facilitation being my personal favourite in this instance given how easily the authors could bring that into the mix) will make for a fantastic addition to the work.
- Given the scope of Nature Communications, the authors should expand their discussion so as to make concrete examples where this quasi-universality scaling can be leveraged in practical applications. This is key given that the entire discussion, as the authors themselves acknowledge, "exclude network-forming liquids, covalent-bonding, strongly ionic and dipolar liquids".

Overall, this is an impressive piece of work: I would be happy to recommend a thoroughly revised version of the manuscript, addressing with sufficient detail the issues I have raised above, for publication in Nature Communication.

Reviewer #1 (Remarks to the Author):

The manuscript reports an extensive numerical study on excess entropy scaling of transport coefficients in supercooled liquid mixtures. The authors show that excess entropy scaling holds in mixtures across compositions and explain this in terms of the "R-simple" nature of the studied systems. Previous work (Ref.22) focused on mixtures of hard spheres or Gaussian particles, which are not R-simple and for which excess entropy scaling does not hold. Additionally, the authors provide some evidence of "quasi-universal" behavior across mixtures with different interactions, that is, diffusion and viscosity collapse on an approximate master curve $f(S_{ex})$, which is a function of the excess entropy S_{ex} alone. The analysis carries over to experimental data for multi-component metallic alloys.

The study is supported by a tremendous numerical effort, which significantly extends previous attempts to assess entropy scaling in liquid mixtures. The paper is also very well written and provides a clear and compact presentation of entropy scaling approaches to the dynamics of supercooled liquids. It is a valuable contribution to the field and may also serve as a dive-in introduction for non-specialists. However, I have two main criticisms and a few minor points that I ask the authors to address before I can recommend publication.

We thank the reviewer for the kind words and the effort put into providing a very careful review. We respond point-by-point below.

1) My main problem is with the notion of "quasi-universality", which I find ambiguous. Several other theoretical approaches to the glass transition predict generic functional forms and scaling relations. Small deviations from collapse can be used to favor one approach over another (cf. Elmatad et al. JPCB 2010) and are interpreted as a failure of the theory. Given the sizable deviations seen in Fig. 3, why should we grant excess entropy scaling the privilege of holding "quasi-universally", as opposed to "failing"? I agree that entropy scaling has no free parameters, still it is not clear to me that these deviations are small enough to claim that the scaling holds "quasi-universally". The authors make no effort to assess whether the observed deviations are small or large in a quantitative sense, nor if these deviations can be accounted for by some generalized scaling.

The word "quasiuniversal" is indeed open for interpretation, which is also shown when Reviewers #2 and #3 find the quasiuniversality established or "hard to dispute", in agreement with our own view on the matter. Based on the rather open interpretation connected to the word, we have decided to remove the word "Quasiuniversal" from the title of the manuscript.

We do acknowledge that there are "sizeable" deviations in the supercooled regime in Fig. 3, but find that such a clear visual trend is observed in both viscosity and diffusion over eight orders of magnitude - for a very wide range of binary mixtures in simulations and experiments - that the quasiuniversal scaling is hard to ignore and demands a theoretical explanation.

We now quantify whether the observed deviations of Fig. 3 are large, or small, by comparing with excess entropy scaling for a small molecule. In Fig. 3b (and corresponding text on page 5) we now compare the binary mixture data with a small dumbbell, almost sphere-like, molecule. The deviations become a factor of 10^5 larger for the deepest supercoolings (this number is extrapolated due to the extremely long simulations of this manuscript that we cannot hope to replicate), ruling out any quasiuniversal behaviour between this molecule and the binary mixtures. For this particular model it is clear that quasi-universality fails which emphasises the above point that the binary mixture case really calls for a theoretical explanation (i.e., is quasiuniversal).

Here is one suggestion to address this point. Suppose you take different mixtures based on inverse power interactions (truly R-simple): would entropy scaling hold exactly across them? If this is the case, one could try to rationalize deviations from collapse in Fig.3 as due to the degradation of pressure-energy correlations. Stated differently, liquids that are more R-simple would collapse more universally than the others. The local pressure-energy correlation coefficient would quantify these deviations. If this were true, one would have a well-defined reference behavior [providing the exact shape of $f(S_{ex})$] and a quantitative measure to account for deviations. From Fig.3 one recognizes for instance that deviations are most pronounced for CuZr, for which the pressure-energy correlation coefficient is lower than 90%.

We thank the reviewer for this suggestion which motivated us to create a new section called “Departure from universality” on page 10 considering this issue in much more detail. For single-component IPLs, and thus also binary IPL mixtures, with different interaction exponents r^{-n} the scaling *amongst them* would not hold exact [see, e.g., Pond et al., JCP 134, 081101 (2011)]. We therefore find it to be more relevant to consider the density-scaling exponent γ , rather than R, in order to try to establish a quantitative measure of deviations.

Interestingly, we find in the new section that the density-scaling exponent is highly correlated to the departure occurring in the supercooled regime. Figure 6 (new) shows that a smaller value for γ moves the curve up for viscosity and down for diffusion. The opposite being the case for larger γ values. More similar γ values, irrespective of the system complexity (three-body, mixing rule, etc), therefore conform to a more similar scaling in the supercooled regime. The new section on page 10 presents this new and important observation and also provides an empirical correction of viscosity based on the value of γ . We have also added text to the discussion on page 11 emphasising the role of γ ; γ does not explain the deviations but does provide a quantitative measure of when the scaling should hold even better (i.e., more similar γ values). This may provide a hint at explaining the deviations in the future.

2) There is an overall ambiguity as to what one really learns from excess entropy scaling. The authors start off in p.1 by saying "This theoretical framework does not in general, however, suggest any causal link among isomorph invariants". However, in the conclusions

they end up saying "a long-standing issue in the study of supercooled liquids is what controls the dynamics. We find that the excess entropy not only controls the viscosity and the diffusion coefficient [...] but also controls them in a quasiuniversal way." Why should excess entropy, as opposed to any other isomorph invariant, "control" the dynamics? Perhaps it's a matter of language, but "controlling" the dynamics implies to me some causal link, which is not proven yet I think. Please clarify.

The isomorph theory states that certain quantities are invariant along the same curves in the thermodynamic phase diagram. One can, equally well, postulate that the relaxation time "controls" the excess entropy rather than the other way round. We agree that a causal link has therefore not been proven in this manuscript, only that the excess entropy correlates very well to the viscosity and diffusion coefficient. We have therefore changed the word "control" to "rationalize" or "correlate" where appropriate in the manuscript.

The concept of causality is fundamental in physics, and is interpreted differently depending on the theory. In relativity the light cone plays an essential role in defining this concept clearly, where only events in the back cone can be considered as causes. What then is causality and control related to the glass transition? Control could be interpreted as: can largely explain or justify the variation in the dynamics, but alas the notion of causality is largely up to what is decided as the independent variable. This is perhaps also why the glass transition is so rich in theories. We tend to look for a variable that doesn't change much upon cooling, but it need not be this way. We have added a short discussion around causality and the isomorph theory on page 11 and for clarity deleted the mentioned sentence on page 1.

Minor points:

- p.3: when presenting the overview of the results for the 2:1 Kob-Andersen mixture, it would be good to remind the T_{MCT} value for this non-canonical composition.

This is now added on page 3. $T_{MCT} = 0.55$ for 2:1 KA.

- p.3 The transition between "Excess entropy scaling" and "Composition-independent excess entropy scaling" sections is a bit awkward. Composition-independent scaling is already shown to hold for diffusion (Fig. 1c and 1d), but the section "Composition-independent excess entropy scaling" starts as this were a new result.

Figures 1c and d show that data can be collapsed as a function of temperature and density for a fixed composition (e.g., 4:1 KA). Figure 2 expands on these results by including also composition as a variable for the same system (e.g., KA), where one would not expect any collapse and therefore it is given special attention in a separate section. To avoid confusion, we have changed the title of that section to "Composition excess entropy scaling" and added a few clarifying words on the same page emphasising that composition scaling is a new result.

- It would be good to clarify the origin of the discrepancy with the findings of Krekelberg et al. in Ref. 22 (which focused on non R-simple mixtures).

This is now added on page 12. Thank you.

- p.5: "A specific value of S_{ex} corresponds to a specific value of D_e due to quasiuniversality, and the breakdown is therefore bound to occur at more or less the same value of D_A or η_e for all systems." and also on p. 9 "As the viscosity and diffusion both show quasiuniversality, their product must also show the same quasiuniversality."

The authors are forcibly ignoring the role of fluctuations here. D and η show quasi-universality essentially by compensation of errors, since the /deviations/ from universality go in opposite directions (opposite signs) for D and η , see Fig. 3.

We do not agree that the quasiuniversality of the SE breakdown arises essentially to a cancellation of errors. It is correct that there are opposite directions for D and η for a given system. However, we now understand that these deviations are linked to γ (see previous discussion), with more similar γ values providing a more similar scaling. This indicates to us that it is not merely a cancellation of errors, but is rather a fundamental observation. We have added a short comment on page 6 expressing the fact of opposite trends for D and η .

- p.5: "We thus conclude that quasiuniversality for binary mixtures explains the observations of Flenner et al. (also p.9)"

The main claim of Flenner et al. was that the features of dynamic heterogeneities are universal in the studied range of temperatures. This is a much stronger claim than "all systems deviate from SE around some S_{ex} value". Please rewrite to clarify the precise connection with Ref.65

We thank the reviewer for bringing our attention to this. The sentence was not referring to dynamic heterogeneities as is the main message of Flenner et al. The above sentence (and other places) has now been rewritten to emphasize that we only talk about the universal SE breakdown observation, and not dynamical heterogeneities. It would be interesting in a future study to also look at dynamic heterogeneities from the excess entropy perspective.

- Related to point 1) above, it would be useful to report the U-W correlation coefficients in the big table in the SI. Also, it would be useful to provide that table in a format suitable for further data analysis, like in csv format.

R and γ values are now added to the SI and csv file format provided in a DOI.

- The authors mention that the longest simulations lasted more than 12 months on GPU. Given the single precision arithmetics used in the force calculation, how could the stability of the simulations be enforced (and how was it assessed) over such a long time?!

The simulations were all run in the NVT ensemble and thus any numerical drift occurring both in single and double precision over these long simulations is thereby eliminated. Crystallisation was checked using order parameters, potential energy, etc. Furthermore, after equilibration (with sanity checks during the equilibration run) it was confirmed that the results are reproducible by running the production-run simulation back-to-back at least twice. We are thus confident on the validity of our long-time simulation results. We have added more of these details to the Method section on page 12.

As an aside, we intend to make starting configurations of these extremely long simulations in RUMD available in a DOI for other researchers to use.

--

Daniele Coslovich

Reviewer #2 (Remarks to the Author):

Report on

Quasiuniversal Excess Entropy Scaling in Supercooled Binary Mixtures

Ian H. Bell, Jeppe C. Dyre, and Trond S. Ingebrigtsen

The present work reports on investigations of the quasiuniversality of transport properties versus excess entropy for a large class of binary mixtures. From the pioneering work of Rosenfeld who first highlight the possibility of a universal excess entropy scaling of transport properties such as diffusion, shear viscosity, and thermal diffusivity, a wealth of subsequent studies were devoted to testing such a law in various situations and systems. Nevertheless, during the three decade history of this still open issue, well described by the authors, very little efforts have been given to the investigation of the possible extension of this scaling law in deep undercooling conditions. And this is precisely the aim of the present work. For this purpose, numerically intensive molecular dynamic simulations for various binary mixtures and several compositions of them were performed in order to cover up to eight orders of magnitude variation of the viscosity for instance in the undercooled region. The authors found a quasiuniversal excess entropy scaling independent of composition and nature of the interactions. Such a finding seems to be correlated to the fact that they can be characterized as R-simple liquids with a correlation coefficient R larger than 0.9. A link with the Breakdown of the Stokes-Einstein relation is then proposed. The conclusions are clear and outcome and further suggestions of research lines are appealing. For all these reasons, I would recommend publication of this paper in Nature Communications. I Suggest the authors to consider the following remarks.

We thank the reviewer for the kind words and the effort put into providing a very careful review. We respond point-by-point below.

1. A significant place in the introduction is given to concept of isomorphism developed by the authors essentially in the last decade. Authors mention that the dimensionless excess entropy is invariant along isomorphs, and as so do the transport coefficients justifies theoretically a priori a universal scaling law `a la Rosenfeld. However, this aspect is not really developed the paper. Probably it would be worth considering this aspect more deeply, as one can realize that the cooling process to reach deep undercooling states are probably not located on isomorphs.

The current study focuses on supercooled state points which have been carefully equilibrated by long simulations. Our results are therefore independent of the cooling process, and each state point, as such, belongs to a particular isomorph. Such isomorphs in supercooled, metastable liquids have been studied and discussed in detail in, e.g., Refs. 40–57 of the current manuscript. We now mention this on page 2.

2. One aspect of the present simulation data is the decoupling of the diffusion coefficients

of the two species, as can be seen in the tables of the supplement materials. This aspect, in relation to the quasiuniversality, has not been developed in the paper. However it seem that this decoupling is related to the local structural and chemical ordering which can be different around each species (see for instance: npj Computational Materials 3, 33, 2017, and related works). The differences in radial-distribution functions shown in Fig. 5 could be a manifestation of that, and might have a more direct effect on the dynamics, and also on the breakdown of the Stokes-Einstein relation. This may deserves a comment and considering both species for the diffusion.

In the previous version of the manuscript, we found in SI that also the (smaller) B-particle shows quasiuniversality when considering its diffusion coefficient, D_B ; see Fig. 2 in SI.

Related to the RDF differences of Fig 5., a previous study [Ref. 95; JCP 143, 044503 (2015)] showed that the locally favored structures (LFS) in the 4:1 KA model is bicapped square antiprisms, but this LFS becomes more compositionally favored in the 2:1 KA mixture. Furthermore, the WS model (usually the 1:1 WS, not the 3:1 we compare with in Fig. 5) favors the formation of Frank-Casper bonds in the supercooled liquid [Phys. Rev. Lett. 104, 105701 (2010)]. The observed differences in the RDFs of Fig. 5 are therefore likely connected to (different) local orderings present in the liquid and between the liquids as suggested. Yet, interestingly, they all show quasiuniversality.

We commented on this intriguing point only in passing on page 8 in the previous version of the manuscript, but have now expanded this discussion on page 8 and 10, adding the reference from above. We acknowledge that the decoupling aspect is very important for quasiuniversality when considering the diffusion coefficient, and have added more text to the discussion on page 12 to make that point clear.

Reviewer #3 (Remarks to the Author):

This manuscript reports computer simulations seeking to demonstrate the emergence and validity of a quasi-universal scaling relation (based on excess entropy) in supercooled binary (simple) mixtures.

This work is very fundamental and touches on one of the long-standing open questions in physical chemistry, i.e. the nature of the glass transition. While it could be argued that the practical relevance of this research is not immediately clear, these sort of advances bear a substantial impact across different fields - I would thus judge this work of broad enough interest to be published in Nature Communications.

To strengthen this point, I would note that the paper is very well written: the introduction is very clear, approaching the problem in a very logical way while referencing most of the relevant literature. The results are presented in a simple and logical way, with no embellishments while making for a truly enjoyable reading.

Onto the technical side of things, the results are robust and the authors took special care in cross-validating them with the existing literature data. Given the diverse array of systems they have investigated, the quasi-universality of the scaling relation they put forward is hard to dispute.

We thank the reviewer for the kind words and the effort put into providing a very careful review. We respond point-by-point below.

I do have a few issues with the current version of the manuscript, though, namely:

- The “quasi-universality” labelling is indeed adequate, given the deviations of ~ 2 orders of magnitude observed in e.g. Fig. 3a at strong supercooling. However the authors failed to provide any explanation whatsoever as to what could be the origin of said deviations: this is a crucial point that deserves further investigation. For instance, the authors could think of potential avenues to exacerbate these deviations - perhaps the R coefficient is playing a role? I feel this is an important piece of the puzzle that truly cripples the impact of this work.

We thank the reviewer for this suggestion which motivated us to create a new section called “Departure from universality” on page 10 considering this issue in much more detail. For single-component IPLs, and thus also binary IPL mixtures, with different interaction exponents r^{-n} the scaling *amongst them* would not hold exact [see, e.g., Pond et al., JCP 134, 081101 (2011)]. We therefore find it to be more relevant to consider the density-scaling exponent γ , rather than R , in order to try to establish a quantitative measure of the deviations.

Interestingly, we find in the new section that the density-scaling exponent is highly correlated to the departure occurring in the supercooled regime. Figure 6 (new) shows that a

lower value of gamma moves the curve up for viscosity and down for diffusion. The opposite being the case for larger gamma values. More similar gamma values, irrespective of the system complexity (three-body, mixing rule, etc), therefore conform to a more similar scaling in the supercooled regime. The new section on page 10 considers this new and important observation and also provides an empirical correction of viscosity based on the value of gamma. We have added new text to the discussion on page 11 emphasising the role of gamma; gamma does not explain the deviations but provides a quantitative measure of when the scaling should hold even better (i.e., more similar gamma values). This may provide a hint at explaining the deviations in the future.

- Using the radial distribution function alone to probe the extent of the validity of the Young-Andersen principle is not enough. At the very least the authors should offer the customary complementary analysis of the distribution of the volumes of the Voronoi polyhedra within the liquids, as the radial distribution function does not uniquely characterise a disordered system. I also believe there is a huge opportunity here to elaborate on the impact of potential structural features on the scaling law put forward by the authors, whom are in fact in a good position to make a connection with what they define as “the nature behind the SE breakdown”. A connection with some of the theories dealing with the nature of the SE breakdown (Dynamical Facilitation being my personal favourite in this instance given how easily the authors could bring that into the mix) will make for a fantastic addition to the work.

The SI now includes a comparison of the Voronoi volume distribution between the different systems corresponding to Fig. 5. We find that the Voronoi distribution is also different amongst the compared systems, being most pronounced for the KA mixtures. This is now mentioned on page 9. This result also indicates that free volume, as such, is not the primary controlling structural parameter.

A connection between fundamental glass transition theories and the observed quasiuniversal SE breakdown is now made on page 7 by comparing the fractional SE exponent obtained from the simulations with predictions obtained from dynamical facilitation (the 1-d East model). Our simulations go 3-4 decades below T_{MCT} , and enables a more careful comparison than ever before in simulations. As such, the excess entropy approach does not provide any predictions for this exponent. We find that the 1-d East model with a (numerically determined) fractional SE exponent of around 0.73 is almost identical to that obtained from our extremely supercooled 2:1 KA data. This result motivates an in-depth study of dynamical facilitation in the context of our 2:1 KA data, which we plan for in a future publication.

- Given the scope of Nature Communications, the authors should expand their discussion so as to make concrete examples where this quasi-universality scaling can be leveraged in practical applications. This is key given that the entire discussion, as the authors themselves acknowledge, “exclude network-forming liquids, covalent-bonding, strongly ionic and dipolar liquids”.

We have expanded the discussion on page 12 describing how the excess entropy scaling may be used to develop novel alloys in the future by providing predictive guidance.

Overall, this is an impressive piece of work: I would be happy to recommend a thoroughly revised version of the manuscript, addressing with sufficient detail the issues I have raised above, for publication in Nature Communication.

We thank all reviewers again for their suggestions, which have greatly improved the manuscript, and are pleased to know that the manuscript was enjoyed.

REVIEWERS' COMMENTS:

Reviewer #1 (Remarks to the Author):

The authors have successfully addressed all the pointed raised in my report. I now recommend publication.

Reviewer #3 (Remarks to the Author):

The Authors have thoroughly addresses all the issues I have raised with respect to the original version of this manuscript. It is quite telling that all the Reviewers identified the "quasi-" issue re: the universality scaling as a potential hurdle: however, the Authors have accumulated additional results which serve very well to clear that sticky point - while improving what was an already excellent piece of work.

I believe the new "Departure from universality" section is absolutely key: while a solid explanation for this behaviour remains to be found (I must admit I am now rather curious at this point...), I do appreciate that these findings alone are not only entirely novel, but of great relevance for the community as they are.

As such, I am happy to recommend the publication of this work in Nature Communications.

One not-so-minor comment which I would appreciate if the authors could address: the fitting coefficients reported on page 10 have no uncertainties associated with them. In fact, they have been reported with a perhaps over-optimistic precision of six decimal digits. While I do not doubt the overall quality of the fit, I do feel that including proper confidence intervals is in order when fitting a 5-parameter model...

REVIEWERS' COMMENTS:

Reviewer #1 (Remarks to the Author):

The authors have successfully addressed all the pointed raised in my report. I now recommend publication.

We thank the reviewer for the kind words.

Reviewer #3 (Remarks to the Author):

The Authors have thoroughly addresses all the issues I have raised with respect to the original version of this manuscript. It is quite telling that all the Reviewers identified the "quasi-" issue re: the universality scaling as a potential hurdle: however, the Authors have accumulated additional results which serve very well to clear that sticky point - while improving what was an already excellent piece of work.

I believe the new "Departure from universality" section is absolutely key: while a solid explanation for this behaviour remains to be found (I must admit I am now rather curious at this point...), I do appreciate that these findings alone are not only entirely novel, but of great relevance for the community as they are.

As such, I am happy to recommend the publication of this work in Nature Communications.

We thank the reviewer for the kind words. It is indeed an interesting connection.

One not-so-minor comment which I would appreciate if the authors could address: the fitting coefficients reported on page 10 have no uncertainties associated with them. In fact, they have been reported with a perhaps over-optimistic precision of six decimal digits. While I do not doubt the overall quality of the fit, I do feel that including proper confidence intervals is in order when fitting a 5-parameter model...

This has now been corrected. Thank you.